# Reasons for Failed Trials of Disease-Modifying Treatments for Alzheimer Disease and Their Contribution in Recent Research

**DOI:** 10.3390/biomedicines7040097

**Published:** 2019-12-09

**Authors:** Konstantina G. Yiannopoulou, Aikaterini I. Anastasiou, Venetia Zachariou, Sygkliti-Henrietta Pelidou

**Affiliations:** 1Memory Center, Neurological Department, Henry Dunant Hospital Center, 107 Mesogeion Avenue, 11526 Athens, Greece; 2Medical School of Athens, National and Kapodistrian University of Athens, 11526 Athens, Greece; aikatianast@gmail.com; 3Icahn School of Medicine at Mount Sinai, Nash family Department of Neurosciences, Department of Pharmacological Sciences, and Friedman Brain Institute, New York, NY 11004, USA; venetia.zachariou@mssm.edu; 4Department of Neurology, University of Ioannina, University Hospital of Ioannina, 45500 Ioannina, Greece; epelidou@cc.uoi.gr

**Keywords:** Alzheimer disease, clinical trial fails, disease-modifying treatments, Alzheimer disease biomarkers, combination treatment, clinical trial designs

## Abstract

Despite all scientific efforts and many protracted and expensive clinical trials, no new drug has been approved by FDA for treatment of Alzheimer disease (AD) since 2003. Indeed, more than 200 investigational programs have failed or have been abandoned in the last decade. The most probable explanations for failures of disease-modifying treatments (DMTs) for AD may include late initiation of treatments during the course of AD development, inappropriate drug dosages, erroneous selection of treatment targets, and mainly an inadequate understanding of the complex pathophysiology of AD, which may necessitate combination treatments rather than monotherapy. Clinical trials’ methodological issues have also been criticized. Drug-development research for AD is aimed to overcome these drawbacks. Preclinical and prodromal AD populations, as well as traditionally investigated populations representing all the clinical stages of AD, are included in recent trials. Systematic use of biomarkers in staging preclinical and prodromal AD and of a single primary outcome in trials of prodromal AD are regularly integrated. The application of amyloid, tau, and neurodegeneration biomarkers, including new biomarkers—such as Tau positron emission tomography, neurofilament light chain (blood and Cerebrospinal fluid (CSF) biomarker of axonal degeneration) and neurogranin (CSF biomarker of synaptic functioning)—to clinical trials allows more precise staging of AD. Additionally, use of Bayesian statistics, modifiable clinical trial designs, and clinical trial simulators enrich the trial methodology. Besides, combination therapy regimens are assessed in clinical trials. The above-mentioned diagnostic and statistical advances, which have been recently integrated in clinical trials, are relevant to the recent failures of studies of disease-modifying treatments. Their experiential rather than theoretical origins may better equip potentially successful drug-development strategies.

## 1. Introduction

Given the fact that AD affects mostly people older than 65 years, the increasing expansion of life span leads to a fast-growing number of patients [1]. Consequently, the research focused on treatments has grown intensively. However, despite all arduous research efforts, at the moment there are no effective treatment options for the disease [2,3]. Indeed, no new drug has been approved by FDA for treatment of AD since 2003, although more than 200 therapeutic agents have been assessed in failed or abandoned investigational programs [4,5]. 

Many explanations for failures of candidate disease-modifying treatments (DMTs) for AD have been proposed. The most prominent include late initiation of treatments during the course of AD development, inappropriate drug dosages, wrong selection of main treatment targets, and mainly an inadequate understanding of the complex pathophysiology of AD [6]. A novel approach to the treatment problems seems to necessitate combination treatments rather than monotherapy [7]. Clinical trials’ methodological issues have also been criticized [4]. 

Drug-development research for AD is aimed to overcome these drawbacks. Preclinical and prodromal AD populations, as well as patients representing all the clinical stages of AD, are included in recent trials [8]. Current guidance provided by the US Food and Drug Administration (FDA) for clinical trials in AD further includes use of biomarkers in staging preclinical and prodromal AD and of a single primary outcome in trials of prodromal AD, and additionally the use of Bayesian statistics and modifiable clinical trial designs [3]. Besides, combination therapy regimens are currently assessed in clinical trials [7].

A search of clinicaltrials.gov from 2012 (accessed September 2019) for phase 3 interventional clinical trials that are “terminated” or “completed” for AD identified all pharmacologic AD trials of all agents that have been recently abandoned. The 2019 annual review of the AD drug development pipeline was also used, as well as the relevant publications in PubMed for the same time frame. All the presented studies on failures of the trials are clinical studies. Animal studies are added only when additional information about any studied agent is needed. 

The goal of this review is to highlight different factors that may contribute to the failure of clinical trials for AD. The review will also summarize how information from failed trials is used to better guide new trials.

## 2. Basic Pathophysiology and Neuropathology of AD

The primary histopathologic lesions of AD are the extracellular amyloid plaques and the intracellular Tau neurofibrillary tangles (NFTs) [9]. The amyloid or senile plaques are constituted chiefly of highly insoluble and proteolysis-resistant peptide fibrils produced by β-amyloid (Aβ) cleavage. Aβ peptides with Aβ38, Aβ40 and Aβ42 as the most common variants are produced after the sequential cleavage of the large precursor protein APP by the two enzymes, β-secretase (BACE1) and γ-secretase. However, Aβ is not formed if APP is first acted upon and cleaved by the enzyme α-secretase instead of β-secretase [10]. According to the ‘amyloid hypothesis’ Aβ production in the brain initiates a cascade of events leading to the clinical syndrome of AD. Aβ is a protein consisting of three main isoforms, Ab38, Aβ40 and Aβ42. Aβ38 and Aβ42 are produced at lower levels, approximately 5–20% of the total Aβ detected in most cells, while the main species generated is Aβ40 that usually constitutes over 50% of total detected Aβ. Aβ42 is the most aggregation-prone form and has the tendency to cluster into oligomers. Oligomers can form Aβ-fibrils that will eventually form amyloid plaques. Aβ40 is somewhat aggregation-prone and it is mostly found in the cerebral vasculature as part of ‘cerebral amyloid angiopathy’. Aβ38 is never found in senile plaques. It is soluble, present in the vasculature of sporadic and familial AD patients [9,10]. It is the forming of amyloid oligomers to which neurotoxicity is mainly attributed and initiates the amyloid cascade. The elements of the cascade include local inflammation, oxidation, excitoxicity (excessive glutamate) and tau hyperphosphorylation [9]. Tau protein is a microtubule-associated protein which binds microtubules in cells to facilitate the neuronal transport system. Microtubules also stabilize growing axons necessary for neuronal development and function. Abnormally hyperphosphorylated tau forms insoluble fibrils and folds into intraneuronic tangles. Consequently, it uncouples from microtubules, inhibits transport and results in microtubule disassembly [10]. Although in the amyloid hypothesis, tau hyperphosphorylation was thought to be a downstream event of Aβ deposition, it is equally probable that tau and Aβ act in parallel pathways causing AD and enhancing each other’s toxic effects [2]. Progressive neuronal destruction leads to shortage and imbalance between various neurotransmitters (e.g., acetylcholine, dopamine, serotonin) and to the cognitive deficiencies seen in AD [9].

Thus, both Aβ and tau are prime targets for DMTs in AD. From this point of view, AD could be prevented or effectively treated by decreasing the production of Aβ and tau; preventing aggregation or misfolding of these proteins; neutralizing or removing the toxic aggregate or misfolded forms of these proteins; or a combination of these modalities [10].

A number of additional pathogenic mechanisms have been described, possibly overlapping with Aβ plaques and NFT formation or induced by them, including inflammation, oxidative damage, iron deregulation and cholesterol metabolism blood–brain barrier dysfunction or α-synuclein toxicity [5,10,11].

## 3. Explanations for Failures of Candidate DMTs for AD and the Consequent Shift in Current Clinical Trials

### 3.1. Inadequate Understanding of the Complex Pathophysiology of AD: Wrong Selection of Main Treatment Target and Inappropriate Drug Dosages

Since lack of efficacy of all agents that were studied in phase 3 trials cannot be accurately explained at this time, it is obvious that the current science is not sufficiently advanced and investigators need to recognize the possibility of deficiency of our knowledge.

Multiple phase 3 failures of agents that aim to reduce beta-amyloid plaques caused researchers to abandon the singular focus on amyloid cascade model. Indeed, if patients with high amyloid levels participate in trials for amyloid clearing drugs and they show no cognitive benefits, it is reasonable to suggest that other or additional pathophysiological substrates need to be targeted [11,12]. 

Recent failures in phase 3 studies of anti-amyloid agents in patients with early stage, mild or mild to moderate AD involved some of the γ-secretase inhibitors, β secretase inhibitors, monoclonal antibodies (mAbs) and intravenous immunoglobulins (IVIg). Additionally, some tau aggregation inhibitors have also failed in phase 3 studies (Table 1).

Inhibitors of γ-Secretase abandoned in phase 3 studies are semagecestat [13], avagacestat [14] and tarenflurbil [15]. Semagecestat was associated with worsening of daily function and increased rates of skin cancer and infection, avagacestat was associated with higher progression rate of the disease and adverse dose-limiting effects (skin cancer), and tarenflurbil was ascribed to low potency and brain penetration.

Further examples of agents targeting Aβ that failed due to lack of efficacy include the β-Secretase (BACE) inhibitors lanabecestat [16], verubecestat [17] and atabecestat [18]. These drugs target the β site amyloid-precursor-protein-cleaving enzyme-1 (BACE-1), and although they demonstrated proof of mechanism of action by lowering the plasma and CSF biomarkers Aβ40 and Aβ42, they failed to prove clinical benefit. The clinical trial of verubecestat in mild to moderate AD was terminated early due to lack of efficacy. A more recent verubecestat trial targeting patients with prodromal AD showed even more disappointing results. Adverse effects, and worsening of cognition and daily function were more common in the verubecestat groups than in the placebo group [17].

Passive Ab immunotherapy via mAbs has been the most active and remains highly promising. A point of concern in these therapies is the occurrence of cerebral microhaemorrhages and vasogenic edema. The underlying mechanism is probably related to vascular amyloid deposits (congophilic amyloid angiopathy), present in nearly all patients with AD. The need for vascular repair and regeneration during Aβ immunotherapy is another argument for early treatment and subtle clearance over a long period of time [9,11,12]. Valuable experience gained from several negative phase 3 trials of the first agents of this class, bapineuzumab [19] and solanezumab [20] paved the way for great insights in mAbs research. Strict inclusion criteria were applied, such as biomarker evidence of AD pathology, specifically “amyloid positivity,” and enrollment of individuals with preclinical stages of the disease. Furthermore, the studies’ design became more specific and targeted: the characteristics of amyloid related imaging abnormalities (ARIA) were associated with antibody dose and APOε4 genotype, necessity for higher dosing and evidence for target engagement (e.g., reduction of plaque burden on amyloid PET) was required [5,21].

Passive Ab immunotherapy via immunoglobulins demonstrated also its own phase 3 failures. Anti-Aβ antibodies are included in naturally occurring autoantibodies. In contrast to mAbs, blood-derived human anti-Aβ immunoglobulin G (IgG) Abs are polyclonal, with lower avidity for single Aβ molecules, and higher for a broader range of epitopes, especially in Aβ oligomers and fibrils. The presence of natural anti-Aβ antibodies have been reported in IV immunoglobulin (IVIg), thus IVIg has been proposed as a potential AD treatment. IVIg is derived from plasma of healthy donors and contains a majority of the human IgG-type antibodies [19,22]. Nevertheless, the first completed phase 3 trial of IVIg for AD, showed good tolerability but lack of efficacy of the agent on cognition or function of participants with mild to moderate AD [22]. In this phase 3, double-blind, placebo-controlled trial, IVIg (Gammagard Liquid; Baxalta, Bannockburn, IL, USA) was administered intravenously (IV) at doses of 0.2 or 0.4 g/kg every two weeks for 18 months.

Besides anti-amyloid agents, tau aggregation inhibitors comprise another category of DMTs that has been tested and initially failed in AD trials. A phenothiazine with tau aggregation inhibition properties, methylene blue (MB), has previously been used in humans and is currently being evaluated in AD trials. MB’s derivative leuco-methylthioninium bis (hydromethanesulphonate) (LMTM) was studied in phase 3 but failed to show a drug placebo difference. Based on the results, a new phase 2/3 trial (LUCIDITY) was started in 2018 in subjects with mild AD with a lower dose of LMTM as monotherapy [23].

In current research a more equipotential conceptualization of AD has been adopted. Amyloid pathology is still targeted, but tau pathology appears to be more firmly associated with early cognitive decline. At the same time, other pathological states such as arteriolosclerosis, aberrant blood–brain barrier, α-synuclein function and synaptic dysfunction are also investigated for their links to AD [11]. Ongoing clinical trials evaluate the efficacy of DMTs amyloid-related mechanisms, Tau-related mechanisms, and DMTs with other mechanisms such as neuroprotection, anti-inflammatory effects, growth factor promotion, metabolic effects, stem cells [3,5].

### 3.2. Inadequate Understanding of the Complex Pathophysiology of AD: Late Initiation of Treatments During the Course of AD Development 

Abnormal deposits of amyloid β and tau tangles and the damage to the brain is believed to start a decade or more before cognitive decline [24].

Lack of efficacy observed in previous phase 3 trials raised the question whether treating AD patients once they become symptomatic may be too late to reverse the progress of neurodegeneration. While Aβ and tau tangles are undetectable at earlier stages, the application of biomarkers for early detection of AD may permit presymptomatic interventions that may halt or delay the progression of the disease [8].

The ongoing development of Aβ agents is a direct result of the previous hypothesis. Although all of the previous agents of this category failed in clinical trials, most of the new agents among them are studied in asymptomatic subjects at risk of developing AD [3,25]. Ongoing clinical trials with active or passive immunotherapy agents [26], with agents that reduce the Aβ plaque burden [27,28], with α-secretase modulators [29] or BACE inhibitors [25] are enrolling prodromal or mild AD patients to test the hypothesis of early pharmacological intervention [3,8].

Hence, the challenge of DMT development for AD has become more complicated as trial populations include also preclinical and prodromal AD, besides AD dementia patients [30]. Accurate classification of stages of AD, especially preclinical stages, demand a new research framework for the diagnosis of AD that may serve clinical trials of AD DMTs [30]. Such a framework based on amyloid, tau, and neurodegeneration biomarkers was introduced by the National Institute on Aging (NIA) and the Alzheimer’s Association [31]. Consequently, most of the current clinical trials have integrated the use of CSF, blood [32] or imaging [33] biomarkers. CSF or blood biomarkers lead the effort to enable more effective DMTs. Their context of use in clinical trials includes patient selection, patient’s classification in a disease state, clarification of therapeutic agent’s mechanism of action, appropriate dose selection and measurement of treatment response [34].

CSF biomarkers that are currently used in clinical trials include Aβ42, total tau (t-tau), and tau phosphorylated at threonine 181 (p-tau) identifying subjects at risk of developing AD [33]. The combination of these biomarkers displays sensitivity of 95% and specificity of 83% in detecting subjects that will develop AD [35], hence it is the main patient-selection tool in trials [32,36].

An ongoing effort for identification of additional biomarkers is remarked. Novel biomarkers could be used for drug-efficiency monitoring, risk classification and prognosis. Several novel biomarkers are considered to be incorporated into drug-development programs [37] (Table 2):

### 3.3. Novel Biomarkers of Aβ Metabolism and Aggregation

CSF Aβ38: Aβ peptides are generated as the result of the sequential cleavage of APP by BACE1 and γ-secretase. The cleavage position of the γ-secretase in the transmembrane domain of APP is not precise, resulting in the production of variable -length Aβ peptides. Aβ42 is already used in clinical trials as a CSF biomarker for AD [33]. Additionally, Aβ peptides shorter than 40 residues have been evaluated for potential utility as AD biomarkers.

CSF Aβ38 has been found to correlate with PET Aβ and the ratio of CSF Aβ42/Aβ38 is better at predicting Aβ-positive PET than CSF Aβ42 alone. Furthermore, CSF Aβ42/Aβ38 may be useful for differentiating between AD and DLB and other non-AD dementias and to detect brain amyloid deposition in prodromal AD and to differentiate AD dementia from non-AD dementias. Aβ42/Aβ38 ratio shows increased accuracy compared to Aβ42 when distinguishing AD from dementia with Lewy bodies or Parkinson’s disease dementia and subcortical vascular dementia, even where all Aβs (including Aβ42) are decreased [38].

CSF Aβ38 has also the promise to be used for patient selection and to demonstrate target engagement of γ-secretase modulators. Commercial assays are already available for this biomarker [32,38].

Plasma BACE1: The main physiological function of BACE1 is APP processing. It is also believed to be a major protease for cell surface proteolysis playing an important role in myelination. Consequently, monitoring of BACE1 activity may be helpful in subjects receiving BACE1 inhibitors. Furthermore, CSF and plasma BACE1 activity was proved to be higher in subjects with mild cognitive impairment (MCI) who progressed to AD compared with those with stable MCI or AD. Plasma BACE1 activity shows ability for prognosis and patient selection. Commercial assays are already available for both BACE1 protein levels and BACE1 activity [39].

### 3.4. Vascular System’s Novel Biomarkers

Vascular dysregulation has been proved to be a contributing factor to AD. Recent work supports that it is also the earliest and strongest pathological factor associated with late-onset AD [40].

CSF and serum heart-type fatty acid-binding protein (hFABP) which has been proposed as a biomarker of myocardial infarction has been identified as potential AD biomarker. Furthermore, it can predict progression from MCI to AD [41], correlates with brain atrophy among individuals with low CSF Aβ42 [42] and differentiates AD from Parkinson’s disease [43]. The source of hFABP in CSF is uncertain but it is highly expressed in the brain. hFABP could help in patient selection and prognosis. Commercial assays are available for hFABP [43].

### 3.5. Novel Biomarkers of Inflammation and Glial Activation

Inflammation is another main contributor to AD pathogenesis. Aβ plaques and NFTs induce an immune response in the brain, which is mediated by activated glial cells. Although the activation of glial cells serves normally to protect the brain, uncontrolled activation can lead to the loss of their homeostatic functions and the acquisition of proinflammation. Thus, reactive oxygen species and nitric oxide are released and contribute to neuronal cell death [32].

CSF and peripheral blood triggering receptor expressed on myeloid cells 2 (TREM2) is expressed by microglial cells in the central nervous system (CNS) and its functions include the regulation of phagocytosis and inflammation. TREM2 expression is upregulated in AD brains, where it protects the brain in the early stages, through the phagocytic clearance of Aβ, but in the later stages, induces activation of the inflammatory response [32]. Higher CSF and peripheral blood TREM2 levels in AD and higher CSF TREM2 levels in MCI groups compared with controls have been observed, supporting possible use in patient selection. Commercial assays are available for this biomarker [44,45].

CSF and blood chitinase-3-like protein 1 (YKL-40) is expressed in astrocytes and Aβ plaques and is connected with inflammation, angiogenesis and tau pathology. CSF YKL-40 levels have been found to be high in AD patients and in the late preclinical AD stages compared with early preclinical stages [32,46]. CSF YKL-40 is regarded as a biomarker of neuroinflammation or astrogliosis in AD and probably can help in patient selection and prognosis. Commercial assays are already available [46].

### 3.6. Novel Biomarkers for Synaptic Dysfunction

Synaptic dysfunction is an early event in AD pathogenesis [32]. The level of synaptic loss in post-mortem brains is correlated with pre-mortem cognitive function in patients with MCI or early AD [32]. It is also found that the synaptic loss in AD is more severe than the neuronal loss in the same cortical region [32].

CSF Neurogranin is mainly found in dendritic spines and in post-synaptic signaling pathways. It is involved in long-term potentiation and memory consolidation [2]. It has been shown to predict disease progression in several studies, even in cognitively normal controls [47,48]. Its levels are correlated with brain atrophy in subjects with Aβ pathology and with rapid cognitive deterioration during clinical follow-up [2]. It is regarded as a potentially useful AD biomarker for patient selection and prognosis. Commercial assays are already available [47].

CSF SNAP-25 and synaptotagmin are synaptic proteins that take part in the mediation of exocytosis of synaptic vesicles for neurotransmitter release. The levels of these proteins are elevated in AD and MCI. They are suggested as potential AD biomarkers for patient selection. Commercial assays are already available for both of them [49].

Synaptic biomarkers could be useful for both prognosis and therapeutic response [2].

### 3.7. Novel Biomarkers for α-Synuclein Pathology

α-Synuclein is a plentiful neuronal protein localized in the presynaptic terminals and involved in vesicle fusion and neurotransmitter release [32]. Aggregates of α-synuclein are intracellular inclusions characteristic of the neurodegenerative diseases termed α-synucleinopathies (Parkinson’s disease, Parkinson’s disease dementia, dementia with Lewy bodies, multiple system atrophy). Nevertheless, α-synuclein aggregates are also found in sporadic AD brains, in Down’s syndrome brains with AD pathology and in familial AD with *PSEN 1* mutations. The relationship between α-synuclein and AD pathology is vague, although many studies suggest that α-synuclein can act synergistically with both Aβ and tau and promotes their aggregation [32].

CSF α-synuclein levels may be useful for identifying Lewy body pathology among AD patients, thus this molecule could be used for patient selection [50].

### 3.8. Novel Biomarkers for TDP-43 Pathology

TDP-43 is a protein capable of binding both DNA and RNA and is involved in transcription and splicing. TDP-43 creates cytoplasmic inclusions observed in amyotrophic lateral sclerosis and in many frontotemporal dementia syndromes. TDP-43 pathology is also detected in 20%–50% of AD patients and is associated with greater brain atrophy and cognitive impairment. The TDP-43 pathology can be triggered by Aβ peptides, and contributes to neuroinflammation, mitochondrial and neural dysfunction [32].

Plasma TDP-43 has been found elevated in AD and in pre-MCI patients who progressed to AD. Since commercial assays are already available, TDP-43 may serve as an AD biomarker for patient selection and prognosis [51].

### 3.9. Iron Metabolism Associated Novel Biomarkers

Excess iron in the brain causes neurodegeneration. It is responsible for the cognitive decline in the genetic disorders classified as neurodegeneration with brain iron accumulation [32]. Elevated iron has been found in AD and MCI brains. Intracellular iron can induce APP processing and induce aggregation of hyperphosphorylated tau [32].

Since ferritin plays a major role in brain iron homeostasis, plasma and CSF ferritin may be used as AD biomarkers. CSF Ferritin may become a prognostic biomarker while plasma ferritin could be used for the screening of preclinical AD. Commercial assays are available for both plasma and CSF ferritin detection [52].

### 3.10. Oxidative Stress Biomarkers

Oxidative stress has been recognized as a mediator of early pathology in AD patients [53]. Reactive oxygen species (ROS) can alter the physical structures of proteins and accompanied by reactive nitrogen species (RNS) can also induce cell membrane lipids to undergo peroxidation under oxidative stress conditions. Altered proteins produce molecules that damage DNA and RNA. All these oxidative stress products accumulate and trigger AD development [54].

Plasma oxidative stress biomarkers associated with MCI and AD are divided into the following categories:Biomarkers associated with damage to proteins: decreased plasma superoxide dismutase (SOD) activity accompanied with increased levels of oxidized proteins has been observed in MCI in comparison to healthy participants (HC). Plasma glutathione reductase/glutathione peroxidase (oxidized proteins) ratio (GR/GPx ratio) also showed statistically significant differences between AD and MCI in a recent study—thus is considered an accumulative biomarker in the disease progression [55].Biomarkers associated with lipid peroxidation: Urine, plasma and CSF 8,12-isoiPF(2alpha)-VI [56] and plasma malondialdehyde (MDA) [45] showed statistically significant differences between AD and MCI patients, and were also considered reliable biomarkers of AD progression. Additionally, some plasma lipid peroxidation compounds (PGF_2α_, isoprostanes, neuroprostanes, isofurans, neurofurans) showed statistically significant correlation with medial temporal atrophy in AD and MCI patients [57].Biomarkers associated with damage to DNA: plasma and CSF 8-hydroxy-2′-deoxyguanosine (8-OHdG) is the most studied biomarker of oxidative DNA damage. Significantly higher levels of this biomarker in AD than in healthy controls (HC) have been observed. Increased levels of 8OHdG and 8-hydroxyguanosine (8OHD) are indicative of DNA and RNA oxidation [58].Total antioxidant capacity determined by the ferric-reducing antioxidant power (FRAP assay) and indirect antioxidants plasma levels (vitamin E, selenium) were decreased in MCI and AD compared to HC, but not yet in a statistically significant and accumulative pattern [59].Others: the APO E genotype has been studied in order to correlate genetic risk factors with oxidative stress biomarkers in AD. E4 allele carriers MCI patients have significantly decreased plasma SOD activity compared to non-E4 carriers, with no further difference for other oxidative-stress biomarkers between the two groups [60].

### 3.11. Other Neuronal Proteins as Novel Biomarkers

CSF Visinin-like protein 1 (VILIP-1) is a neuronal calcium-sensor protein related to synaptic plasticity in signaling pathways, which is abundantly produced in the brain [32]. CSF VILIP-1 levels have been proved to be elevated in AD patients in many studies and may be used as prognostic biomarker of incipient cognitive decline, of cognitive decline rates and brain atrophy rates, of progression from MCI to AD, and of AD differentiation from other dementias [61]. VILIP-1 commercial assays are already available [61].

CSF and plasma NF-L (neurofilament lights) are promising biomarkers. NF-Ls are expressed in neurons and particularly in axons. They are partly responsible for the transmission of electrical impulses and for normal synaptic function [62]. Abnormal aggregation of neurofilaments are evident in several neurological diseases including AD [62]. NF-L subunit is known to be increased in many neurodegenerative diseases, supporting its role as a marker of axonal injury [62].

CSF NF-L levels have been shown to be elevated in AD and MCI patients and to have a linear correlation with cognitive impairment and survival time in AD patients [61]. Plasma NF-L have been found to be increased in pre-symptomatic subjects known to be carriers of AD-causing gene mutations and patients with MCI or AD [63,64]. They seem also to be correlated with brain atrophy [65]. CSF NF-L could be useful as biomarkers for prognosis, and plasma NF-L could be useful as a non-invasive biomarker for patient selection and prognosis. Commercial and in vitro assays are available [32].

In addition to CSF and blood biomarkers, imaging biomarkers contribute to stratification of patients in disease stages and to the evaluation of disease progression in DMT clinical trials even in the absence of noticeable cognitive impairment [33]. Volumetric MRI, T1 and T2-weighted MRI are useful in structural imaging and quantitative analysis of atrophy in MCI and AD patients. The hippocampus and entorhinal cortex may be the first regions affected by atrophy in MCI and AD. Functional MRIs are used for detecting disease-specific alterations in cognitive functions. Structural and functional MRI findings can predict AD onset in patients with MCI [66,67,68]. Amyloid positron emission tomography (PET) is used for detecting the amyloid deposits in preclinical AD and MCI patients and for monitoring the progressive amyloid burden in the brain [69].

The most promising imaging biomarkers seem to be the tau-targeted PET tracers such as fluortaucipir, which are explored in numerous studies [70,71]. The advent of these tracers enables researchers to investigate the sequence of accumulation of tau and Aβ in correlation with age and with development of cognitive impairment due to AD. Recent results show that elevated Flortaucipir tau binding is associated with an increased prevalence of cognitive impairment and support further evaluation of tau PET imaging as a possible biomarker for diagnosis, patient staging, and monitoring effects in AD DMT clinical trials [72].

Blood and CSF biomarkers, along with imaging and elaborate memory-scale measurements may be combined to generate disease signatures, and a better categorization of patients on the basis of AD stage and severity. This information will better guide the selection of suitable patients, and eventually the development tailored and efficacious treatment approaches.

#### 3.11.1. Inadequate Understanding of the Complex Pathophysiology of AD: Combination Treatments

Due to the complex pathophysiology of AD, patients may remain unresponsive to monotherapy treatments. Notably all single-agent DMTs have failed to halt the disease progression. Consequently, combination treatments may be necessary to delay or halt the cognitive and functional deterioration by the disease [33].

Combination trials are different from add-on trials, which are typically used for new therapies in AD, and they compare a new agent to placebo in patients who are already receiving a background treatment. In combination trials two drugs are assessed separately, in combination, and in comparison with placebo in a 2 × 2 trial design. Using this methodology, investigators can better assess the synergistic and individual effects of each drug [7]. The main benefit of this method is that two or more mediators of the disease can be simultaneously targeted (e.g., amyloid and tau). This approach also permits interventions in a single target (e.g., amyloid) by two complementary mechanisms of action [33].

Two combination DMTs are currently in phase 3 clinical trials: (a) The ALZT-OPT1 clinical trial in patients with early AD, assessing cromolyn, an antiamyloid regimen in combination with the anti-inflammatory ibuprofen [73]. (b) The second phase 3 combination trial for gamunex, delivers human albumin through plasma exchange in combination with infusion of intravenous immunoglobulin [7]. This trial targets amyloid by two mechanisms: First, by removing and replacing albumin bound to pathogenic elements of Aβ that cross the blood brain barrier by plasma exchange. This action permits further transfer of Aβ out of the central nervous system. The second mechanism involves actions of intravenous immunoglobulin, which may further increase the clearance of amyloid [74].

The multiple challenges of treating AD and the complex pathophysiology of this disorder have led therapeutic efforts towards multi-agent approaches. Such approaches permit efficient interventions in multiple pathways or interventions in different components of the same pathway. Multi-targeting drugs have been successfully used in other diseases, including HIV and cancer therapeutics [75].

On the other hand, combination treatments option does not seem to be a panacea. One outstanding exception in this trend is the low dose leuco-methylthioninium bis (hydromethanesulphonate) (LMTM) monotherapy for treatment of mild AD, which is currently assessed in a phase 3 clinical trial [23]. The 100 mg twice a day as monotherapy subgroup was compared to 4 mg twice a day as randomized, and the 4 mg twice a day as monotherapy subgroup was compared to the 4 mg twice a day as add-on therapy in the standard symptomatic treatments for AD (cholinesterase inhibitors and/or memantine) subgroup. The results supported the hypothesis that LMTM might be more effective as monotherapy and that 4 mg twice a day may serve as well as higher doses. Consequently, only low-dose LMTM is tested in phase 3 [23].

#### 3.11.2. Methodological Issues

In addition to the proposed reasons for the failures of trials of DMTs for AD disease-modifying drugs discussed above, issues with clinical trial design and methodology should also be considered [4,76]. Indeed, new innovative study designs are currently applied.

In placebo-controlled 2 × 2 trial designs, patients are randomized to agent A plus placebo, agent B plus placebo, A plus B and placebo plus placebo [77]. This type of study design is used in combination treatment trials and in add-on studies with some modifications, enabling the simultaneous assessment of agents used alone or in combination.

The 3-arm study design, is also used in current trials. In this case patients are randomized to treatment agent A, B, or A plus B [7]. The benefits of this design are that no patient remains untreated and that smaller samples of patients are required. However, this design is less appropriate for later stages of drug development, when every different aspect of every drug must be assessed [7].

Furthermore, several adaptive trials are under way for AD. Adaptive endpoints include drug effects on cognitive and clinical measures, a dose-escalation algorithm, novel imaging biomarkers, early disease core and novel biomarkers and later-disease cognitive assessments [78]. Most of novel trials use an adaptive bayesian design to predict effectiveness or failure of individual agents and they adaptively randomize patients to the most efficient drugs [78]. Additionally, the interim analyses permit termination of the study when a predefined signal is detected, therefore accelerating decision making. Adaptive randomization and interim analyses can reduce the size and duration of the trial and they may prevent advancing to phase 3 in cases that data show no clinical efficacy [79].

The most impressive and influential case of correction of a methodological issue is that of aducanumab trials [80]. Aducanumab is an anti-Aβ oligomers monoclonal antibody which has been studied in two phase 3 efficacy trials: The 221AD301 ENGAGE study with 1350 people with MCI due to AD or mild AD and the identical 221AD302 EMERGE study conducted in 1350 additional patients. An interim analysis predicted that EMERGE and ENGAGE would miss their primary endpoints, thus aducanumab was removed from any further study.

On 22 October 2019, it was announced that the interim futility analysis of aducanumab studies was wrong. On the contrary, the subsequent analysis of a larger data showed that EMERGE had met its primary endpoint. Specifically, patients receiving the highest dose of 10 mg/kg, had a significant reduction in decline on CDR-SB, the primary endpoint. The same group also showed a lower decline on secondary endpoints (ADCS-ADL-MCI, MMSE, ADAS-Cog). Subsequently, an exploratory analysis was conducted in the ENGAGE trial which also did not meet the primary endpoint and suggested that the same subgroup of participants declined less and more slowly.

An application for regulatory approval for aducanumab in the FDA is planned for early 2020 [81].

## 4. Discussion and Conclusions

Given the complexity of AD and the high DMT failure rate, the treatment of AD patients remains challenging. The complex pathologic pathways of AD in combination with our incomplete understanding of the relationships among the numerous mechanisms involved in the pathophysiology and progress of the disease seem to be mainly responsible for the failure of clinical trials. The amyloid hypothesis itself has clearly great influence over the direction of DMT clinical trials but this approach has so far proven totally unsuccessful. Targeting of NFT deposits is not yet fully explored but also no encouraging results have been met so far. Furthermore, there is no evidence that modifying the levels of considered AD biomarkers, such as Aβ and tau, predicts clinical benefit [82].

We dare say that amyloid plaques and NFT deposits might be just a usual pathologic finding in AD, in the same way that a tubercle is characteristic of tuberculosis. We mean that these formations are probably a result of other causative factors or a reaction to them.

The last call for the amyloid hypothesis is probably use of monoclonal antibodies directed against Aβ oligomers. According to the updated version of the Aβ cascade hypothesis of AD, oligomers represent the major pathogenic species of Aβ. Aducanumab (BIIB037) and BAN2401 (mAb158), anti-Aβ oligomers monoclonal antibodies, have shown positive signals of clinical efficacy [83].

To sum up, we propose the following future directions to consider for AD research moving forward:Although recent evidence still supports the possibility that Aβ status could predict AD risk and play a significant role in disease progression, it does query the perceived centrality of its role in the causation or definition of the disease. Consequently, we suggest that future clinical research should not always have to be approached through an Aβ lens.Besides Aβ biomarkers, other possible biomarkers, such as biomarkers of inflammation and glial activation, synaptic dysfunction, α-synuclein pathology, TDP-43 pathology, iron metabolism and oxidative stress must be further investigated. Selected blood and CSF biomarkers, along with imaging and elaborate memory scale measurements may be combined to generate disease signatures, and they could also be used for drug efficiency monitoring, risk classification and prognosis. Most of them are considered to be incorporated into drug development programs.The complex pathophysiology of AD probably demands therapeutic efforts towards multi-agent approaches. Such combination DMTs might permit efficient interventions in multiple pathways or interventions in different components of the same pathway.Issues with clinical trial design and methodology are also important. Indeed, new innovative study designs are applied. Adaptive randomization and interim analyses can reduce the size and duration of any trial and they may prevent advancing to phase 3 in cases that data show no clinical efficacy.

## Figures and Tables

**Table 1 biomedicines-07-00097-t001:** Agents that failed in phase 3.

Agent	Agent Mechanism Class	Mechanism of Action	Therapeutic Purpose	N	Parameter Evaluates	Results	Reasons behind Stopping Trial
Semagecestat	Antiamyloid	γ-secretase inhibitor	Reduce amyloid production	463	ADAS-cog ADCS-ADL	No efficacy	worsening of daily function, increased rates of skin cancer and infection
Avagacestat	Antiamyloid	γ-secretase inhibitor	Reduce amyloid production	263	CSF biomarkers amyloid PET ADAS-cog ADCS-ADL	No efficacy	higher progression rate of the disease, skin cancer
Tarenflurbil	Antiamyloid	γ-secretase inhibitor	Reduce amyloid production	1046	ADAS-cog ADCS-ADL	No efficacy	low brain penetration
Lanabecestat	Antiamyloid	BACE1 inhibitor	Reduce amyloid production	1722	ADAS-cog13	No efficacy	futility
Verubecestat	Antiamyloid	BACE1 inhibitor	Reduce amyloid production	1454	CDR-SB	No efficacy	cognition and daily function worsening
Atabecestat	Antiamyloid	BACE1 inhibitor	Reduce amyloid production	18	Ab CSF and Plasma	No efficacy	-
Bapineuzumab	Antiamyloid	Monoclonal antibody directed at plaque and oligomers	Remove amyloid	683 ApoE4 carriers 329 non carriers	Ab and pTau in CSF	No efficacy	Brain edema or effusion, futility
Solanezumab	Antiamyloid	Monoclonal antibody directed at plaque and oligomers	Remove amyloid	2129	ADAS-cog14	No efficacy	futility
Gammagard Liquid (IVIg)	Antiamyloid	Human normal immunoglobulin	Remove amyloid	390	ADAS-cog11 ADCS-ADL	No efficacy	No efficacy
LMTM	Anti-tau	Tau protein aggregation inhibitor	Reduce neurofibrillary tangle formation	891	ADAS-cog	No efficacy	No efficacy

ADAS-cog: Alzheimer’s Disease Assessment Scale for cognition; ADCS-ADL: Alzheimer’s Disease Cooperative Study-Activities of Daily Living scale; CDR-SB: Clinical Dementia Rating Sum of Boxes.

**Table 2 biomedicines-07-00097-t002:** Candidate cerebrospinal fluid (CSF)-plasma biomarkers for Alzheimer disease.

	Biomarker	Utility in AD
Aβ metabolism and aggregation biomarkers		
	✓ CSF Aβ38	patient selection
	✓ Plasma BACE1	patient selection and prognosis
Vascular system biomarkers		
	✓ hFABP (CSF, serum)	patient selection and prognosis
Inflammation and glial activation biomarkers		
	✓ TREM2	increased levels in AD
	✓ YKL-40 (known as CHI3L1)	patient selection; prognostic marker
Synaptic dysfunction biomarkers		
	✓ Neurogranin (CSF)	Disease progression; patient selection and prognosis
	✓ CSF SNAP-25	patient selection
	✓ Synaptotagmin	patient selection
α-Synuclein pathology biomarkers		
	✓ CSF α-synuclein	patient selection
TDP-43 pathology biomarkers		
	✓ Plasma TDP-43	associated with greater brain atrophy and cognitive impairment
Iron metabolism biomarkers		
	✓ Plasma/CSF Ferritin	screening of preclinical AD and prognostic biomarker
Other neuronal protein biomarkers		
	✓ CSF VILIP-1	Early AD diagnosis; differentiating AD-MCA; prognostic biomarker
	✓ CSF/plasma NF-L	biomarkers for prognosis
Oxidative biomarkers		
associated with damage to proteins	✓ SOD (plasma)	differentiating MCI from HC significantly decreased in MCI E4 allele carriers compared to non-E4 carriers
associated with damage to proteins	✓ GR/GPx ratio (plasma)	accumulative biomarker in the disease progression
associated with lipid peroxidation	✓ 8,12-isoiPF(2alpha)-VI (urine, plasma and CSF)	differentiating AD-MCA; biomarkers of AD progression
associated with lipid peroxidation	✓ MDA (plasma)	differentiating AD-MCA; biomarkers of AD progression
associated with damage to DNA	✓ 8-OHdG (plasma, CSF)	differentiating AD from HC
total antioxidant capacity	✓ FRAP assay (plasma)	decreased in MCI and AD compared to HC
indirect antioxidants	✓ vitamin E, selenium (plasma)	decreased in MCI and AD compared to HC

CSF; cerebrospinal fluid, BACE1; β-site amyloid precursor protein cleaving enzyme 1, hFABP; heart-type fatty acid-binding protein, TREM2; triggering receptor expressed on myeloid cells, YKL-40; chitinase-3-like protein 1 (CHI3L1), SNAP-25; synaptosomal-associated protein of 25 kDa, TDP-43; TAR DNA binding protein-43, VILIP-1; visinin-like protein 1, NF-L; neurofilament light. SOD: superoxide dismutase; HC, healthy controls; GR/GPx ratio, glutathione reductase/glutathione peroxidase ratio; MDA: malondialdehyde; 8-OHdG: 8-hydroxy-2′-deoxyguanosine; FRAP assay: ferric reducing antioxidant power.

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
