# Peer review of "Reasons for Failed Trials of Disease-Modifying Treatments for Alzheimer Disease and Their Contribution in Recent Research"

_biomedicines, 2019, doi:10.3390/biomedicines7040097_

Round 1

Reviewer 1 Report

Yiannopoulou et al. summarize different Alzheimer’s clinical trials that have failed. This is an interesting review that could enhance our understanding of many reasons that are responsible for the failures of many Alzheimer’s clinical trials. This manuscript is well written with a good standard of English. The manuscript is worthy of consideration for publication though a few questions need to be addressed first.

The authors highlighted many fluids AD biomarkers, however, they did not discuss the oxidative stress as a biomarker for AD. A significant amount of evidence has shown that oxidative stress is an important pathogenic feature in AD. Oxidative damages on neuronal lipids and proteins are important feature of AD. The author should highlight the most important biomarkers of oxidative stress in AD.

The authors mentioned that combination of treatments rather than monotherapy may be necessary to delay or halt AD, however, one of the current drug in research that targets tau protein is LMTM, it is currently in Phase 3 clinical trial and the data so far showed that LMTM might be effective as monotherapy rather than in combination with the standard symptomatic treatments for Alzheimer’s disease (cholinesterase inhibitors and/or memantine), (see Wilcock et al 2018; Potential of Low Dose Leuco-Methylthioninium is (Hydromethanesulphonate) (LMTM) Monotherapy for Treatment of Mild Alzheimer’s Disease: Cohort Analysis as Modified Primary Outcome in a Phase III Clinical Trial ). I think the author should discuss this work and include it in the review.

Author Response

The authors highlighted many fluids AD biomarkers, however, they did not discuss the oxidative stress as a biomarker for AD. A significant amount of evidence has shown that oxidative stress is an important pathogenic feature in AD. Oxidative damages on neuronal lipids and proteins are important feature of AD. The author should highlight the most important biomarkers of oxidative stress in AD.

Biomarkers of oxidative stress should be definitely added and we are grateful to the reviewer for his/her contribution to the thoroughness of our review article.

We added this information in red in the manuscript:

Oxidative stress biomarkers

Many clinical trials and animal studies have recognized oxidative stress as mediator of early injury in AD patients and AD models [43]. Under oxidative stress conditions reactive oxygen species (ROS) and reactive nitrogen species (RNS) can induce peroxidation of cell membrane lipids while ROS can also damage proteins altering their physical structures. Protein damaged products contribute to indirect damage to DNA and RNA. The accumulation of all this oxidative stress products can trigger AD development [44].

Plasma oxidative stress biomarkers associated with MCI due to AD are divided into the following categories:

compounds associated with damage to proteins: increased levels of oxidized proteins and decreased plasma superoxide dismutase (SOD) activity have been observed in MCI relative to HC. In one study the plasma glutathione reductase/glutathione peroxidase (oxidized proteins) ratio (GR/GPx ratio)showed also statistically significant differences between AD and MCI, thus it is considered an accumulative biomarker in the disease progression [45].

compounds associated with lipid peroxidation: 8,12-isoiPF(2alpha)-VIin urine, plasma and CSF [46] and plasma malondialdehyde (MDA)[45] showed statistically significant differences between AD and MCI groups and were also considered reliable biomarkers of AD progression. Additionally, some lipid peroxidation compounds (neuroprostanes, isoprostanes, neurofurans, isofurans, 17-epi-17-F2t-dihomo-IsoP, PGF2α) in plasma showed statistically significant correlation with medial temporal atrophy [47]. compounds associated with damage to DNA: 8-hydroxy-2′-deoxyguanosine (8-OHdG)is the most studied biomarker of oxidative DNA damage in plasma, CSF and brain tissue. Significantly higher levels of this biomarker in AD than in healthy participants (HC) have been found. Increased levels of 8OHdG and 8-hydroxyguanosine (8OHD) indicate DNA and RNA oxidation [48]. antioxidants (total antioxidant capacity determined by the ferric reducing antioxidant power (FRAP assay) and indirect antioxidants plasma levels (vitamin E, selenium)were found decreased in MCI and AD compared to HC, but not yet in a statistically significant and accumulative pattern [49]. others: APO E genotype has been studied in order to correlate genetic risk factors with oxidative stress biomarkers in AD . E4 allelecarriers MCI patients have showed significantly decreased plasma superoxide dismutase activity compared to to non-E4 carriers, with no further difference for oxidative stress biomarkers between the two groups [50].

The authors mentioned that combination of treatments rather than monotherapy may be necessary to delay or halt AD, however, one of the current drug in research that targets tau protein is LMTM, it is currently in Phase 3 clinical trial and the data so far showed that LMTM might be effective as monotherapy rather than in combination with the standard symptomatic treatments for Alzheimer’s disease (cholinesterase inhibitors and/or memantine), (see Wilcock et al 2018; Potential of Low Dose Leuco-Methylthioninium is (Hydromethanesulphonate) (LMTM) Monotherapy for Treatment of Mild Alzheimer’s Disease: Cohort Analysis as Modified Primary Outcome in a Phase III Clinical Trial ). I think the author should discuss this work and include it in the review. 

We also added the appropriate information about LMTM monotherapy as it was highlighted by the reviewer, in red fonts too:

 On the other hand, combination treatments option does not seem to be a panacea. One outstanding exception in this trend is  the Low Dose Leuco-Methylthioninium Bis (Hydromethanesulphonate) (LMTM) Monotherapy for Treatment of Mild Alzheimer's Disease, which is currently assessed in Phase 3 clinical trial [68]. Prior to unblinding, the statistical analysis plan was revised to compare the 100 mg twice a day as monotherapy subgroup versus 4 mg twice a day as randomized, and 4 mg twice a day as monotherapy versus 4 mg twice a day as add-on therapy in the standard symptomatic treatments for AD (cholinesterase inhibitors and/or memantine). The results supported the hypothesis that LMTM might be effective as monotherapy and that 4 mg twice a day may serve as well as higher doses, thus only low dose LMTM is tested in Phase 3.

Reviewer 2 Report

The manuscript raised a burning issue and address the reasons behind disease-modifying treatments for Alzheimer Disease. However, there are few major comments in the manuscripts which need to be addressed:

Page 2; Line 87-90: Sentence doesn't give a clear outcome, look like orphan sentence. Need to rewrite the sentence.

Page 3; Line 96: That's true, however add the sentence which will mention about early investigation of amyloid beta and tau tangles are not detected in the body. Discuss about the solution of early detection and their biomarkers.

Page 3; Line 125: The given biomarkers of Aβ metabolism and aggregation are detected at what stage? early Alzheimer's, MCI and late Alzheimer's.

Page 3; Line 126: Add a sentence about CSF Aβ38 significance over Aβ40, and Aβ42.

Page 5; Line 224: It is highly important to select the patient on the basis of severity of Alzheimer's disease and stages. Now the question started how to diagnose the patients- Based on CSF and Blood Biomarkers? Memory scale measurement? radio imaging or PET imaging? These sections should be addressed which may determine the selection of suitable patients.

Page 6; Line 241: There is a need to discuss about each above section in terms of their feasibility, stages and significance and conclude the paper with future direction.

Author Response

Page 2; Line 87-90: Sentence doesn't give a clear outcome, look like orphan sentence. Need to rewrite the sentence:

We tried to improve the whole paragraph, which appears in bold yellow in the manuscript:

 In current research a more equipotential conceptualization of AD has been adopted. Amyloid pathology is still targeted, but tau pathology appears to be more firmly associated with early cognitive decline. At the same time, other pathological states such as arteriolosclerosis, aberrant blood–brain barrier or α-synuclein function are also investigated for their links to AD [11]. Ongoing clinical trials evaluate the efficacy of DMTs amyloid-related mechanisms, Tau-related mechanisms, and DMTs with other mechanisms such as neuroprotection, anti-inflammatory effects, growth factor promotion, metabolic effects, stem cell therapies [3,5].

 Page 3; Line 96: That's true, however add the sentence which will mention about early investigation of amyloid beta and tau tangles are not detected in the body. Discuss about the solution of early detection and their biomarkers.

We added the necessary sentence which was suggested by our reviewer,  in boldyellow fonts in the manuscript.

 The lack of efficacy observed in previous phase 3 trials raised the question whether treating AD patients once they become symptomatic may be too late to reverse the progress of neurodegeneration. While beta amyloid and tau tangles are undetectable at earlier stages, the application of biomarkers for early detection of AD may permit presymptomatic interventions that may halt or delay the progression of the disease [8].

 Page 3; Line 125: The given biomarkers of Aβmetabolism and aggregation are detected at what stage? early Alzheimer's, MCI and late Alzheimer's. Page 3; Line 126: Add a sentence about CSF Aβ38 significance over Aβ40, and Aβ

The whole section about biomarkers is rewritten and every new component is added in blue fonts in the manuscript. We added the information asked by our reviewer about biomarkers of Ab metabolism and aggregation and Aβ38 significance over Aβ40:

CSF Aβ38:  Aβpeptides are generated as the result of the sequential cleavage of amyloid precursor protein (APP) by  BACE1 and γ-secretase. The cleavage position of the γ-secretase in the transmembrane domain of APP is not precise, resulting in the production of variable length Aβpeptides.  Aβ42 is  already used in clinical trials as a CSF biomarker for AD [33]. Additionally, Aβpeptides shorter than 40 residues have been evaluated for potential utility as AD biomarkers.

CSF Aβ38 has been found to correlate with PET Aβand the ratio of CSF Aβ42/ Aβ38 is better at predicting Aβ-positive PET than CSF Aβ42 alone. Furthermore, CSF Aβ42/Aβ38 may be useful for differentiating between AD and DLB and other non-AD dementias and to detect brain amyloid deposition in prodromal AD. Aβ42/Aβ38 ratio shows increased accuracy compared to Aβ42 when distinguishing AD from dementia with Lewy bodies or Parkinson's disease dementia and subcortical vascular dementia, even where all Aβs (including Aβ42) are decreased [38].

CSF Aβ38 has also the promise to be used for patient selection and to demonstrate target engagement of γ-secretase modulators. Commercial assays are already available for this biomarker [32, 38].

 Plasma BACE1: The main physiological function of BACE1 is APP processing. It is also believed to be a major protease for cell surface proteolysis playing an important role in myelination. Consequently, monitoring of BACE1 activity may be helpful in subjects receiving BACE1 inhibitors. Furthermore, CSF and plasma BACE1 activity was proved to be higher in subjects with MCI who progressed to AD compared with those with stable MCI or AD. Plasma BACE1 activity shows ability for prognosis and patient selection. Commercial assays are already available for both BACE1 protein levels and BACE1 activity [39].

Page 5; Line 224: It is highly important to select the patient on the basis of severity of Alzheimer's disease and stages. Now the question started how to diagnose the patients- Based on CSF and Blood Biomarkers? Memory scale measurement? radio imaging or PET imaging? These sections should be addressed which may determine the selection of suitable patients.

This question is answered in the section  b. Inadequate understanding of the complex pathophysiology of AD: late initiation of treatments during the course of AD development:  

Hence, the challenge of DMT development for AD has become more complicated as trial populations  include also preclinical and prodromal AD, besides AD dementia patients [30]. Accurate classification of stages of AD, especially preclinical stages, demand a new research framework for the diagnosis of AD that may serve clinical trials of DMTs in AD [30]. Such a framework based on amyloid, tau, and neurodegeneration biomarkers was introduced by the National Institute on Aging (NIA) and the Alzheimer’s Association [31]. Consequently, most of the current clinical trials have integrated the use of fluid [32] or imaging [33] biomarkers.  Cerebrospinal fluid (CSF) or blood fluid biomarkers lead the effort to enable more effective DMTs development in AD. Their context of use in clinical trials includes patient selection, patient’s classification in a disease state, clarification of therapeutic agent’s mechanism of action,  appropriate dose selection and measurement of treatment response [34].

 Page 6; Line 241: There is a need to discuss about each above section in terms of their feasibility, stages and significance and conclude the paper with future direction.

We tried to address our reviewer’s indications in the conclusion section (in bold yellow)

Discussion - Conclusion

Given the complexity of AD and the high failure rate of the DMTs in development, the treatment of AD patients remains challenging. The complex pathologic pathways of AD in combination with our incomplete understanding of the relationships among the numerous mechanisms involved in the pathophysiology and progress of the disease seem to be mainly responsible for the failure of clinical trials. Targeting well selected or multiple pathologic pathways, earlier initiation of treatment, integration of fluid and imaging biomarkers for patient selection, prognosis, monitoring treatment efficacy, as well as implementation of new innovative study designs are fundamental for the development of tailored and effective treatments. While it may not be feasible to use all available imaging or molecular tools to obtain a “disease signature” in the clinical practice, the application of multiple markers and image analysis data in patients selected for clinical trials is expected better dissect patient populations and expedite the evaluation of novel treatments. Meanwhile, research efforts on AD substrates and markers along with more sensitive imaging tools may provide new directions toward the diagnosis and early treatment of this neurodegenerative disorder.

 Reviewer 3 Report

Questions:

1. Line 65 “Since lack of efficacy of all agents that were studied in phase 3 trials cannot be accurately explained at this time,” please describe phase 3 trials because there no any other background of phase 3 trial.

2. I understand authors must be knowledgeable about Alzheimer Disease, but there still need some backgrounds of AD, like clinical biomarker, neuropathology, general pathway of AD. Or summarize all of these informations as graphic description. It could help public to understand the paper well.

3. For “Novel fluid biomarkers of Ab metabolism and aggregation”, author listed all biomarkers simply, but didn’t discuss the connection of them and AD phathology.

4. Line 215 “by combining plasma exchange with binding (bound?) amyloid.” What’s mean about (bound?)?

The paper is a review, which could help others understand studies and researches systematically so far about AD. But the paper lack necessary backgrouds and information. The paper could be accepted after major revision or add more systematic data.

Author Response

Line 65 “Since lack of efficacy of all agents that were studied in phase 3 trials cannot be accurately explained at this time,” please describe phase 3 trials because there no any other background of phase 3 trial.

We added a description of failed phase 3 trials (in blue fonts) and a Table (Table 1. in green fonts) in an effort to address our reviewer’s 3 justified indications:

Recent  failures in phase 3 studies of anti- amyloid agents in patients with early stage, mild or mild to moderate Alzheimer’s disease involved some of the γ-secretase inhibitors, β secretase inhibitors, monoclonal Antibodies (mAbs) and intravenous immunoglobulins (IVIg). Additionally, sometau aggregation inhibitorshave also failed in phase 3 studies.

γ-secretase inhibitors abandoned in phase 3 studies are Semagecestat [11],Avagacestat [12] and Tarenflurbil [13]. Semagecestat was associated with worsening of daily function and increased rates of skin cancer and infection, Avagacestat was associated with higher progression rate of the disease and adverse dose-limiting effects (skin cancer)and Tarenflurbil was ascribed to low potency and brain penetration.

Further examples of agents targeting beta- amyloid that failed in phase 3 due to lack of efficacy include the β secretase inhibitors (BACE)  Lanabecestat [14], Verubecestat [15] and Atabecestat [16].  These drugs target the β site amyloid-precursor-protein-cleaving enzyme-1 (BACE-1), and although they demonstrated proof of mechanism of action by lowering the plasma and CSF biomarkers Aβ40 and Aβ42, they failed to prove clinical benefit. The clinical trial of Verubecestat in mild to moderate AD was terminated early due to lack of efficacy. A more recent Verubecestat trial targeting patients with prodromal AD showed even more disappointing results. Adverse events, cognition and daily function worsening were more common in the Verubecestat groups than in the placebo group [15].

Passive Ab immunotherapy via mAbs has been the most active class of agents and remains highly promising. A point of concern in these therapies is the occurrence of cerebral microhemorrhages and vasogenic edema. The underlying mechanism is probably related to vascular amyloid deposits (congophilic amyloid angiopathy), present in nearly all patients with AD. The need for vascular repair and regeneration during Aβimmunotherapy is another argument for early treatment and subtle clearance over a long period of time [17]. Valuable experience gained from several negative phase 3 trials of the first agents of this class, bapineuzumab [18] and solanezumab [19] paved the way for great insights in mAbs research. Strict inclusion criteria were applied, such as biomarker evidence of AD pathology, specifically “amyloid positivity,” and enrollment of individuals with preclinical stages of the disease. Furthermore, the studies’ design became more specific and targeted: the characteristics of amyloid related imaging abnormalities (ARIA) were associated with antibody dose and APOε4 genotype, necessity for higher dosing was recognized and evidence for target engagement (e.g., reduction of plaque burden on amyloid PET) was required [5].

Additionally, passive Ab immunotherapy via immunoglobulins demonstrated also its own phase 3 failures.Anti-Aβantibodies are included in naturally occurring autoantibodies. In contrast to mAbs, blood-derived human anti-Aβimmunoglobulin G (IgG) Abs are polyclonal, with lower avidity for single Aβmolecules, and higher for a broader range of epitopes, especially in Aβoligomers and fibrils. The presence of natural anti-Aβantibodies have been reported in IV immunoglobulin (IVIg), thus IVIg has been proposed as a potential AD treatment. IVIg is derived from plasma of healthy donors and contains a majority of the human IgG-type antibodies [17,20]. Nevertheless, the first completed phase 3 trial of IVIg for AD, showed good tolerability but lack of efficacy of the agent on cognition or function of participants with mild to moderate AD [21]. In this phase 3, double-blind, placebo-controlled trial, IVIg (Gammagard Liquid; Baxalta, Bannockburn, IL) was administered IV at doses of 0.2 or 0.4 g/kg every 2 weeks for 18 months. 

Besides anti-amyloid agents, tau aggregation inhibitors is another category of DMTs that has been tested and initially failed in AD trials. A phenothiazine with tau aggregation inhibition properties, methylene blue (MB), has previously been used in humans and is currently being evaluated in AD trials. MB’s derivative Leuco-Methylthioninium Bis(Hydromethanesulphonate) (LMTM) was studied in phase 3 but failed to show a drug-placebo difference. Based on the results, a new phase 2/3 trial (LUCIDITY) was started in 2018 in subjects with mild AD with a lower dose of LMTM as monotherapy [22].

Table 1. Agents that failed in phase 3.

Agent

Agent mechanism class

Mechanism of action

Therapeutic purpose

N

parameter evaluates

results

reasons behind stopping trial

Semagecestat

Antiamyloid

γ – secretase inhibitor

Reduce amyloid production

463

ADAS-cog

ADCS-ADL

No efficacy

worsening of daily function,

increased rates of skin cancer and infection

Avagacestat

Antiamyloid

γ – secretase inhibitor

Reduce amyloid production

263

CSF biomarkers amyloid PET

ADAS-cog

ADCS-ADL

No efficacy

higher progression rate of the disease,

skin cancer

Tarenflurbil

Antiamyloid

γ – secretase inhibitor

Reduce amyloid production

1046 

ADAS-cog

ADCS-ADL

No efficacy

low brain penetration.

Lanabecestat

Antiamyloid

BACE1inhibitor

Reduce amyloid production

1722

ADAS-cog

13

No efficacy

futility

Verubecestat

Antiamyloid

BACE1 inhibitor

Reduce amyloid production

1454 

CDR-SB

No efficacy

cognition and daily function worsening

Atabecestat

Antiamyloid

BACE1 inhibitor

Reduce amyloid production

18

Ab CSF and Plasma

No efficacy

-

Bapineuzumab

Antiamyloid

Monoclonal antibody directed at plaque and oligomers

Remove amyloid

683

ApoE4 carriers

329

non carriers

Ab and pTau in CSF

No efficacy

Brain edema

or effusion,

futility

solanezumab

Antiamyloid

Monoclonal antibody directed at plaque and oligomers

Remove amyloid

2129

ADAS-cog14

No efficacy

futility

Gammagard Liquid (IVIg)

Antiamyloid

human normal immunoglobulin

Remove amyloid

390 

ADAS-cog11

ADCS-ADL

No efficacy

No efficacy

LMTM

Anti-tau

Tau protein aggregation inhibitor

Reduce neurofibrillary tangle formation

891

ADAS-cog

No efficacy

No efficacy

ADAS-cog: Alzheimer's Disease Assessment Scale for cognition

ADCS-ADL: Alzheimer's Disease Cooperative Study-Activities of Daily Living scale

CDR – SB: Clinical Dementia Rating Sum of Boxes

2. I understand authors must be knowledgeable about Alzheimer Disease, but there still need some backgrounds of AD, like clinical biomarker, neuropathology, general pathway of AD. Or summarize all of these informations as graphic description. It could help public to understand the paper well.

We added a new section about Basic pathophysiology and neuropathology of AD (in blue fonts in the manuscript)

 Basic pathophysiology and neuropathology of AD

The primary histopathologic lesions of AD are the extracellular amyloid (Aβ) plaques and the intracellular Tau neurofibrillary tangles (NFTs)[8].The Aβor senile plaques are highly insoluble and proteolysis-resistant fibrils that constitute chiefly of the neurotoxic peptide Aβ, which is produced after the sequential cleavage of the large precursor protein APP by the two enzymes, β-secretase (BACE1) and γ-secretase. However, Aβis not formed if APP is first acted upon and cleaved by the enzyme α-secretase instead of β-secretase[9]. According to the ‘amyloid hypothesis’ Aβproduction in the brain initiates a cascade of events leading to the clinical syndrome of AD. Aβis a protein consisting of two main forms, Aβ40 and Aβ42. Aβ42 is the most soluble form and has the tendency to cluster into oligomers. Oligomers can form Aβ-fibrils that will eventually form amyloid plaques. Aβ40 is mostly found in the cerebral vasculature as part of ‘cerebral amyloid angiopathy’. It is the forming of amyloid oligomers to which neurotoxicity is attributed and initiates the amyloid cascade. The elements of the cascade include local inflammation, oxidation, excitoxicity (excessive glutamate) and tau hyperphosphorylation [8]. Tau protein isa microtubule associated protein which binds microtubules in cells to facilitate the neuronal transport system. Microtubules also stabilize growing axons necessary for the development and growth of neurons Abnormally hyperphosphorylated tau forms insoluble fibrils and folds into intraneuronic tangles. Consequently, it uncouples from microtubules, inhibits transport and results in microtubule disassembly[9]. Although in the amyloid hypothesis, tau hyperphosphorylation was thought to be a downstream event of Aβdeposition, it is equally probable that tau and Aβact in parallel pathways causing AD and enhancing each other’s toxic effects[10].Progressive neuronal destruction leads to shortage and imbalance between various neurotransmitters (e.g. acetylcholine, dopamine, serotonin) and to the cognitive deficiencies seen in AD [8].

Thus, both Aβand tau are prime targets for DMTs in AD. From this point of view, AD could be prevented or effectively treated by decreasing the production of Aβand tau; preventing aggregation or misfolding of these proteins; neutralizing or removing the toxic aggregate or misfolded forms of these proteins; or a combination of these modalities [9].

A number of additional pathogenic mechanisms have been described, possibly overlapping with Aβplaques and NFT formation or induced by them, including inflammation, oxidative damage, iron deregulation and cholesterol metabolism [5,9].

For “Novel fluid biomarkers of Ab metabolism and aggregation”, author listed all biomarkers simply, but didn’t discuss the connection of them and AD phathology.

In order to address our reviewer’s indication we rewrote the biomarkers section adding the needed information in blue and the total new paragraph about oxidative stress biomarkers in red. We also added Table 2.Candidate fluid (CSF-Plasma) biomarkers for Alzheimer’s disease in green.

 Novel fluid biomarkers of Aβmetabolism and aggregation

CSF Aβ38:  Aβpeptides are generated as the result of the sequential cleavage of amyloid precursor protein (APP) by  BACE1 and γ-secretase. The cleavage position of the γ-secretase in the transmembrane domain of APP is not precise, resulting in the production of variable length Aβpeptides.  Aβ42 is  already used in clinical trials as a CSF biomarker for AD [33]. Additionally, Aβpeptides shorter than 40 residues have been evaluated for potential utility as AD biomarkers.

CSF Aβ38 has been found to correlate with PET Aβand the ratio of CSF Aβ42/ Aβ38 is better at predicting Aβ-positive PET than CSF Aβ42 alone. Furthermore, CSF Aβ42/Aβ38 may be useful for differentiating between AD and DLB and other non-AD dementias.

CSF Aβ38 has also the promise to be used for patient selection and to demonstrate target engagement of γ-secretase modulators. Commercial assays are already available for this biomarker [32, 38].

 Plasma BACE1: The main physiological function of BACE1 is APP processing. It is also believed to be a major protease for cell surface proteolysisplaying an important rolein myelination.Consequently, monitoring of BACE1 activity may be helpful in subjects receiving BACE1 inhibitors. Furthermore, CSF and plasma BACE1 activity was proved to be higher in subjects with MCI who progressed to AD compared with those with stable MCI or AD. Plasma BACE1 activity shows ability for prognosis and patient selection. Commercial assays are already available for both BACE1 protein levels and BACE1 activity [39].

-Vascular system’s novel fluid biomarkers

Vascular dysregulation has been proved to be  a contributing factor to AD. Recent work supports that it is also the earliest and strongest pathological factor associated with late-onset AD [40].

CSF and serum Heart-type fatty acid-binding protein ( hFABP)which has been proposed as a biomarker of myocardial infarction has been also identified as a potential AD biomarker. Furthermore, it can predict progression from MCI to AD [41], correlates with brain atrophy among individuals with low CSF Aβ42 [42] and differentiates AD from Parkinson’s disease [43]. The source of hFABP in CSF is uncertain but it is highly expressed in the brain. hFABP could attribute to patient selection and prognosis. Commercial assays are available for hFABP [43].

Novel fluid biomarkers of inflammation and glial activation

Inflammation is another main contributor to AD pathogenesis. Aβplaques and NFTs induce an immune response in the brain, which is mediated by activated glial cells. Although the activation of glial cells serves normally to protect the brain, uncontrolled activation can lead to the loss of their homeostatic functions and the acquisition of proinflammatory skills. Thus,  reactive oxygen species and nitric oxide are released and contribute to neuronal cell death [32].

CSF and peripheral blood Triggering receptor expressed on myeloid cells 2 (TREM2) is expressed by microglial cells in the CNSand its functions includethe regulation of phagocytosisand inflammation. TREM2 expression is upregulated in AD brains, where it protects the brain in the early stages, through the phagocytic clearance of Aβ, but in the later stages, induces activation of the inflammatory response [32]. Higher CSF and peripheral bloodTREM2 levels in AD and higher CSF TREM2 levels in MCI groups compared with controlshave been observed, supporting possible use in patient selection. Commercial assays are available for the measurement of this biomarker [44, 45].

CSF and blood chitinase-3-like protein 1 (YKL-40)is expressed in astrocytes besides Aβ plaques and is connected with inflammation, angiogenesis andtau pathology. CSF YKL-40 levels have been found higher in AD patients compared with controls and in the late preclinical AD stages compared with early preclinical stages [32, 46]. CSF YKL-40 is regarded as a biomarker of neuroinflammation or astrogliosis in AD and probably can help in patient selection and prognosis. Commercial assays are already available [46].

Novel fluid biomarkers for synaptic dysfunction

Synaptic dysfunction is an early event in AD pathogenesis [32]. The level of synaptic loss in post-mortem brains is correlated with pre-mortem cognitive function in patients with MCI or early AD [ 32]. It is also found that the synaptic loss in AD is more severe than the neuronal loss in the same cortical region [32].

CSF Neurograninis mainly found in dendritic spines and its function is expressed in post-synaptic signaling pathways. It is involved in long-term potentiation and memory consolidation [2]. It has been shown to predict disease progression in several studies, even in cognitively normal controls [47, 48]. Its levels are correlated with brain atrophy in subjects with Aβ pathologyand with rapid cognitive deterioration during clinical follow- up [2].It is regarded that it could be useful as an AD biomarker for patient selection and prognosis. Commercial assays are already available [47].

CSF SNAP25 and synaptotagmin are synaptic proteins that take part inthe mediation of exocytosis of synaptic vesicles for neurotransmitter release. The levels of these proteins are elevated in  AD and MCI. They are suggested as potential AD biomarkers for patient selection. Commercial assays are already available for both of them [49].

Synaptic biomarkers could be useful for both prognosis and therapeutic response [2].

Novel fluid biomarkers for α‐Synuclein pathology 

α-Synuclein is an plentiful neuronal protein localized in the presynaptic terminals and involved in vesicle fusion and neurotransmitter release [32]. Aggregates of α-synuclein are intracellular inclusions characteristic of the neurodegenerative diseases termed α-synucleinopathies (Parkinson’s disease,  Parkinson’s disease dementia, dementia with Lewy bodies, multiple system atrophy). Nevertheless, α-synuclein aggregates are also found in sporadic AD brains, in Down’s syndrome brains with AD pathology and in familial AD with PSEN 1 mutations. The relationship between α-synuclein and AD pathology is vague, although many studies suggest that α-synuclein can act synergistically with both Aβand tau and promotes their aggregation[32].

CSF α-synucleinlevels may  be useful for identifying Lewy Body pathology among AD patients, thus this molecule could be used for patient selection [50].

Novel fluid biomarkers for TDP‐43 pathology

TDP-43 is a protein capable of binding both DNA and RNA and is involved in transcription and splicing. TDP-43 creates cytoplasmic inclusions observed in amyotrophic lateral sclerosis and in many frontotemporal dementia syndromes.  TDP-43 pathology is also detected in 20–50% of AD patients is associated with greater brain atrophyand cog- nitive impairment. TDP-43 pathology can be triggered by Aβpeptides,and contributes to neuroinflammation, mitochondrial and neural dysfunction [32].

Plasma TDP43 has been found elevated in AD and in pre-MCI patients who progressed to AD. Since commercial assays are already available, TDP‐43  may serve as an AD biomarker for patient selection and prognosis [51].

Iron metabolism associated novel fluid biomarkers

Iron causes neurodegeneration, when present in excess in the brain. It is responsible for the cognitive decline  in the genetic disorders classified as neurodegeneration with brain iron accumulation [32]. Elevated iron has been found in AD and MCI brains. Intracellular iron can induce APP processing and induce aggregation of hyperphosphorylated tau [32].

Since Ferritin plays a major role in brain iron homeostasis,plasma and CSF Ferritinmay be used as AD biomarkers. CSF Ferritin may become a prognostic biomarker while plasma ferritin could be used for the screening of preclinical AD. Commercial assays are available for both plasma and CSF Ferritin detection [52].

Oxidative stress biomarkers

Oxidative stress has been recognized by many clinical trials as mediator of early pathology in AD patients [53]. Reactive oxygen species (ROS) can alter the physical structures of proteins and accompanied by reactive nitrogen species (RNS) can also induce peroxidation of cell membrane lipids under oxidative stress conditions. Altered proteins produce molecules that damage DNA and RNA. All these oxidative stress products accumulate and trigger AD development [54].

Plasma oxidative stress biomarkers associated with MCI and AD are divided into the following categories:

biomarkers associated with damage to proteins: decreased plasma superoxide dismutase (SOD) activity accompanied with increased levels of oxidized proteins has been observed in MCI in comparison to healthy participants (HC). Plasma glutathione reductase/glutathione peroxidase (oxidized proteins) ratio (GR/GPx ratio)showed also statistically significant differences between AD and MCI in a recent study, thus it is considered an accumulative biomarker in the disease progression [55]. biomarkers associated with lipid peroxidation: Urine, plasma and CSF8,12-isoiPF(2alpha)-VI[56] and plasma malondialdehyde (MDA)[45] showed statistically significant differences between AD and MCI patients and were also considered reliable biomarkers of AD progression. Additionally, some plasma lipid peroxidation compounds (PGF2α,, isoprostanes, neuroprostanes, isofurans, neurofurans) showed statistically significant correlation with medial temporal atrophy in AD and MCI patients [57]. biomarkers associated with damage to DNA: Plasma and CSF 8-hydroxy-2′-deoxyguanosine (8-OHdG)is the most studied biomarker of oxidative DNA damage. Significantly higher levels of this biomarker in AD than in HC have been observed. Increased levels of 8OHdG and 8-hydroxyguanosine (8OHD) are indicative of DNA and RNA oxidation [58]. Total antioxidant capacity determined by the ferric reducing antioxidant power (FRAP assay) and indirect antioxidants plasma levels (vitamin E, selenium)were found decreased in MCI and AD compared to HC, but not yet in a statistically significant and accumulative pattern [59]. others: APO E genotype has been studied in order to correlate genetic risk factors with oxidative stress biomarkers in AD. E4 allelecarriers MCI patients have showed significantly decreased plasma SODactivity compared to to non-E4 carriers, with no further difference for other oxidative stress biomarkers between the two groups [60].

Other neuronal proteins as novel fluid biomarkers

CSF Visinin-like protein 1 (VILIP-1)is a neuronal calcium sensor protein related to synaptic plasticity in signaling pathways, which isabundantly produced in the brain[32]. CSF VILIP-1 levels have been proved to be elevated in AD patients in many studies and may be used as prognostic biomarker of incipient cognitive decline, of cognitive decline’s and brain atrophy’s rates, of progression from MCI to AD and of  AD differentiation from other dementias [61]. VILIP-1 Commercial assays are already available [61].

CSF and plasma NFL (Neurofilament lights) are promising biomarkers. NF-Ls are  expressed in neurons and  particularly  in axons, where are partly responsible for the transmission of electrical impulses and for normal synaptic function [62]. Abnormal aggregation of neurofilaments are evident in several neurological diseases including AD[62].NF-L subunit is known to be increased in many neurodegenerative diseases, supporting its role as a marker of axonal injury [62].

CSF NF-L levels have been shown to be elevated in AD and MCI patients and to have a linear correlation with cognitive impairment and survival time in AD patients [61]. Plasma NF-L have been found to be increased in pre-symptomatic subjects known to be carriers of AD-causing gene mutations and patients with MCI or AD [63, 64]. They seem also to be correlated with brain atrophy [65]. CSF NF-L could be useful as biomarkers for prognosis, and plasma NF-L could be useful as a non-invasive biomarker for patient selection and prognosis. Commercial and in vitro assays are available [66].

4.

Table 2. Candidate fluid (CSF-Plasma) biomarkers for Alzheimer’s disease

Biomarker                 

Utility in Alzheimer disease

Aβ metabolism and aggregation biomarkers

ü  CSF Aβ38  

patient selection

ü  Plasma BACE1

patient selection & prognosis

Vascular system biomarkers

ü  hFABP (CSF, serum)

patient selection & prognosis

Inflammation and glial activation biomarkers

ü  TREM2  

increased levels in Alzheimer’s disease

ü  YKL-40 (known as CHI3L1 )

patient selection; prognostic marker

Synaptic dysfunction biomarkers

ü  Neurogranin (CSF)

Disease progression; patient selection & prognosis

ü  CSF SNAP25

patient selection

ü  synaptotagmin

patient selection

αSynuclein pathology biomarkers

ü  CSF α-synuclein

patient selection

TDP43 pathology biomarkers

ü  Plasma TDP43

Iron metabolism biomarkers

ü  Plasma/CSF Ferritin

screening of preclinical AD and prognostic biomarker

Other neuronal protein biomarkers

ü  CSF VILIP-1

Early AD diagnosis; differentiating AD-MCA; prognostic biomarker

ü  CSF /plasma NFL

biomarkers for prognosis

Oxidative biomarkers

associated with damage to proteins

ü  SOD

differentiating MCI from HC

significantly decreased in MCI E4 allele carriers compared to non-E4 carriers

associated with damage to proteins

ü  GR/GPx ratio

accumulative biomarker in the disease progression

associated with lipid peroxidation

ü  8,12-isoiPF(2alpha)-VI

differentiating AD-MCA;

biomarkers of AD progression

associated with lipid peroxidation

ü  MDA

differentiating AD-MCA;

biomarkers of AD progression

associated with damage to DNA

ü  8-OHdG

differentiating AD from HC

associated with damage to proteins

ü  GR/GPx ratio

accumulative biomarker in the disease progression

 total antioxidant capacity

ü  FRAP assay

decreased in MCI and AD compared to HC

indirect antioxidants

ü  vitamin E, selenium

decreased in MCI and AD compared to HC

CSF; cerebrospinal fluid, BACE1;β-site amyloid precursor protein cleaving enzyme 1, hFABP;Heart-type fatty acid-binding protein, TREM2; Triggering receptor expressed on myeloid cells, YKL-40;Chitinase-3-like protein 1 (CHI3L1), SNAP‐25;synaptosomal-associated protein of 25 kDa, TDP‐43;TAR DNA binding protein-43, VILIP-1; Visinin-like protein 1, NF‐L; Neurofilament light.SOD: superoxide dismutase; HC: Healthy Controls;GR/GPx ratio: glutathione reductase/glutathione peroxidase ratio; MDA: malondialdehyde; 8-OHdG:8-hydroxy-2′-deoxyguanosine; FRAP assay: ferric reducing antioxidant power

Line 215 “by combining plasma exchange with binding (bound?) amyloid.” What’s mean about (bound?)?

We rewrote the phrase with blue fonts in the manuscript:

Second, intravenous immunoglobulin may further increase amyloid clearance by combining plasma exchange with the binding properties of immunoglobulinwith Aβ oligomers and fibrils [76].

Reviewer 4 Report

Yiannopoulou and the group have summarized the reasons for failed clinical trials of treatments for Alzheimer's Disease and lessons learned from the research. The topic is new and not been extensively summarized in the past, so can be accepted for publication. The authors discussed reasons for failures of clinical candidate compounds such as inadequate understanding of pathophysiology of AD, its complex nature, novel fluid markers, use of drug combinations, etc. The review is well written, but I feel it can be expanded with more detail and summarizing important clinical trials in tabular format with agents, outcomes, number of patients, parameter evaluates and results, reasons behind stopping trial will be good. Biomarkers can be also represented in the table format. Authors can include some more references.

Author Response

The review is well written, but I feel it can be expanded with more detail and summarizing important clinical trials in tabular format with agents, outcomes, number of patients, parameter evaluates and results, reasons behind stopping trial will be good.

We expanded the failed clinical trials section ( in blue fonts in the manuscript) and we also added a relevant Table (Table 1. In green).

Recent  failures in phase 3 studies of anti- amyloid agents in patients with early stage, mild or mild to moderate Alzheimer’s disease involved some of the γ-secretase inhibitors, β secretase inhibitors, monoclonal Antibodies (mAbs) and intravenous immunoglobulins (IVIg). Additionally, sometau aggregation inhibitorshave also failed in phase 3 studies.

γ-secretase inhibitors abandoned in phase 3 studies are Semagecestat [13],Avagacestat [14] and Tarenflurbil [15]. Semagecestat was associated with worsening of daily function and increased rates of skin cancer and infection, Avagacestat was associated with higher progression rate of the disease and adverse dose-limiting effects (skin cancer)and Tarenflurbil was ascribed to low potency and brain penetration.

Further examples of agents targeting beta- amyloid that failed in phase 3 due to lack of efficacy include the β secretase inhibitors (BACE-1)  Lanabecestat [16], Verubecestat [17] and Atabecestat [18].  These drugs target the β site amyloid-precursor-protein-cleaving enzyme-1 (BACE-1), and although they demonstrated proof of mechanism of action by lowering the plasma and CSF biomarkers Aβ40 and Aβ42, they failed to prove clinical benefit. The clinical trial of Verubecestat in mild to moderate AD was terminated early due to lack of efficacy. A more recent Verubecestat trial targeting patients with prodromal AD showed even more disappointing results. Adverse events, cognition and daily function worsening were more common in the Verubecestat groups than in the placebo group [17].

Passive Ab immunotherapy via mAbs has been the most active class of agents and remains highly promising. A point of concern in these therapies is the occurrence of cerebral microhemorrhages and vasogenic edema. The underlying mechanism is probably related to vascular amyloid deposits (congophilic amyloid angiopathy), present in nearly all patients with AD. The need for vascular repair and regeneration during Aβimmunotherapy is another argument for early treatment and subtle clearance over a long period of time [9, 11, 12]. Valuable experience gained from several negative phase 3 trials of the first agents of this class, bapineuzumab [19] and solanezumab [20] paved the way for great insights in mAbs research. Strict inclusion criteria were applied, such as biomarker evidence of AD pathology, specifically “amyloid positivity,” and enrollment of individuals with preclinical stages of the disease. Furthermore, the studies’ design became more specific and targeted: the characteristics of amyloid related imaging abnormalities (ARIA) were associated with antibody dose and APOε4 genotype, necessity for higher dosing was recognized and evidence for target engagement (e.g., reduction of plaque burden on amyloid PET) was required [5, 21].

Additionally, passive Ab immunotherapy via immunoglobulins demonstrated also its own phase 3 failures.Anti-Aβantibodies are included in naturally occurring autoantibodies. In contrast to mAbs, blood-derived human anti-Aβimmunoglobulin G (IgG) Abs are polyclonal, with lower avidity for single Aβmolecules, and higher for a broader range of epitopes, especially in Aβoligomers and fibrils. The presence of natural anti-Aβantibodies have been reported in IV immunoglobulin (IVIg), thus IVIg has been proposed as a potential AD treatment. IVIg is derived from plasma of healthy donors and contains a majority of the human IgG-type antibodies [19,22]. Nevertheless, the first completed phase 3 trial of IVIg for AD, showed good tolerability but lack of efficacy of the agent on cognition or function of participants with mild to moderate AD [22]. In this phase 3, double-blind, placebo-controlled trial, IVIg (Gammagard Liquid; Baxalta, Bannockburn, IL) was administered IV at doses of 0.2 or 0.4 g/kg every 2 weeks for 18 months. 

Besides anti-amyloid agents, tau aggregation inhibitors is another category of DMTs that has been tested and initially failed in AD trials. A phenothiazine with tau aggregation inhibition properties, methylene blue (MB), has previously been used in humans and is currently being evaluated in AD trials. MB’s derivative Leuco-Methylthioninium Bis (Hydromethanesulphonate) (LMTM) was studied in phase 3 but failed to show a drug-placebo difference. Based on the results, a new phase 2/3 trial (LUCIDITY) was started in 2018 in subjects with mild AD with a lower dose of LMTM as monotherapy [23].

Table 1. Agents that failed in phase 3.

Agent

Agent mechanism class

Mechanism of action

Therapeutic purpose

N

parameter evaluates

results

reasons behind stopping trial

Semagecestat

Antiamyloid

γ – secretase inhibitor

Reduce amyloid production

463

ADAS-cog

ADCS-ADL

No efficacy

worsening of daily function,

increased rates of skin cancer and infection

Avagacestat

Antiamyloid

γ – secretase inhibitor

Reduce amyloid production

263

CSF biomarkers amyloid PET

ADAS-cog

ADCS-ADL

No efficacy

higher progression rate of the disease,

skin cancer

Tarenflurbil

Antiamyloid

γ – secretase inhibitor

Reduce amyloid production

1046 

ADAS-cog

ADCS-ADL

No efficacy

low brain penetration.

Lanabecestat

Antiamyloid

BACE1inhibitor

Reduce amyloid production

1722

ADAS-cog

13

No efficacy

futility

Verubecestat

Antiamyloid

BACE1 inhibitor

Reduce amyloid production

1454 

CDR-SB

No efficacy

cognition and daily function worsening

Atabecestat

Antiamyloid

BACE1 inhibitor

Reduce amyloid production

18

Ab CSF and Plasma

No efficacy

-

bapineuzumab

Antiamyloid

Monoclonal antibody directed at plaque and oligomers

Remove amyloid

683

ApoE4 carriers

329

non carriers

Ab and pTau in CSF

No efficacy

Brain edema

or effusion,

futility

solanezumab

Antiamyloid

Monoclonal antibody directed at plaque and oligomers

Remove amyloid

2129

ADAS-cog14

No efficacy

futility

Gammagard Liquid (IVIg)

Antiamyloid

human normal immunoglobulin

Remove amyloid

390 

ADAS-cog11

ADCS-ADL

No efficacy

No efficacy

LMTM

Anti-tau

Tau protein aggregation inhibitor

Reduce neurofibrillary tangle formation

891

ADAS-cog

No efficacy

No efficacy

ADAS-cog: Alzheimer's Disease Assessment Scale for cognition

ADCS-ADL: Alzheimer's Disease Cooperative Study-Activities of Daily Living scale

CDR – SB: Clinical Dementia Rating Sum of Boxes

Biomarkers can be also represented in the table format. We added table 2. in green.

Table 2. Candidate fluid (CSF-Plasma) biomarkers for Alzheimer’s disease

Biomarker                 

Utility in Alzheimer disease

Aβ metabolism and aggregation biomarkers

ü  CSF Aβ38  

patient selection

ü  Plasma BACE1

patient selection & prognosis

Vascular system biomarkers

ü  hFABP (CSF, serum)

patient selection & prognosis

Inflammation and glial activation biomarkers

ü  TREM2  

increased levels in Alzheimer’s disease

ü  YKL-40 (known as CHI3L1 )

patient selection; prognostic marker

Synaptic dysfunction biomarkers

ü  Neurogranin (CSF)

Disease progression; patient selection & prognosis

ü  CSF SNAP25

patient selection

ü  synaptotagmin

patient selection

αSynuclein pathology biomarkers

ü  CSF α-synuclein

patient selection

TDP43 pathology biomarkers

ü  Plasma TDP43

Iron metabolism biomarkers

ü  Plasma/CSF Ferritin

screening of preclinical AD and prognostic biomarker

Other neuronal protein biomarkers

ü  CSF VILIP-1

Early AD diagnosis; differentiating AD-MCA; prognostic biomarker

ü  CSF /plasma NFL

biomarkers for prognosis

Oxidative biomarkers

associated with damage to proteins

ü  SOD

differentiating MCI from HC

significantly decreased in MCI E4 allele carriers compared to non-E4 carriers

associated with damage to proteins

ü  GR/GPx ratio

accumulative biomarker in the disease progression

associated with lipid peroxidation

ü  8,12-isoiPF(2alpha)-VI

differentiating AD-MCA;

biomarkers of AD progression

associated with lipid peroxidation

ü  MDA

differentiating AD-MCA;

biomarkers of AD progression

associated with damage to DNA

ü  8-OHdG

differentiating AD from HC

associated with damage to proteins

ü  GR/GPx ratio

accumulative biomarker in the disease progression

 total antioxidant capacity

ü  FRAP assay

decreased in MCI and AD compared to HC

indirect antioxidants

ü  vitamin E, selenium

decreased in MCI and AD compared to HC

CSF; cerebrospinal fluid, BACE1;β-site amyloid precursor protein cleaving enzyme 1, hFABP;Heart-type fatty acid-binding protein, TREM2; Triggering receptor expressed on myeloid cells, YKL-40;Chitinase-3-like protein 1 (CHI3L1), SNAP‐25;synaptosomal-associated protein of 25 kDa, TDP‐43;TAR DNA binding protein-43, VILIP-1; Visinin-like protein 1, NF‐L; Neurofilament light.SOD: superoxide dismutase; HC: Healthy Controls;GR/GPx ratio: glutathione reductase/glutathione peroxidase ratio; MDA: malondialdehyde; 8-OHdG:8-hydroxy-2′-deoxyguanosine; FRAP assay: ferric reducing antioxidant power

Authors can include some more references.

Many more references are included.

Round 2

Reviewer 2 Report

The manuscript improved from the previous version, however still lacking 2 major issues to include and organise properly.

What methods authors have been used to collect the information on failure of trials in Alzheimer's disease, such as what kind of databases and years of the study. The given studies on failure of the trials unable to distinguish between the animal studies and clinical studies.    

Author Response

What methods authors have been used to collect the information on failure of trials in Alzheimer's disease, such as what kind of databases and years of the study. The given studies on failure of the trials unable to distinguish between the animal studies and clinical studies.     

The following paragraph is added in the final manuscript in dark blue underlined font in the introduction section.

A search of clinicaltrials.gov from 2012 (accessed September, 2019) for phase 3  interventional clinical trials that are “terminated” or “ completed” for AD identified all pharmacologic AD trials of all agents that have been  recently abandoned. The 2019 annual review of the AD drug development pipeline was also used for double checking of our findings, as well as the relevant publications in PubMed data base for the same time frame. All the presented studies on failures of the trials are clinical studies. Animal studies are added only when additional information about the studied agent are needed. 

given studies on failure of the trials

Doody, R.S.; Raman, R.; Farlow, M.; Iwatsubo, T.; Vellas, B.; Joffe, S.; Kieburtz, K.; He, F.; Sun, X.; Thomas, R.G.; Aisen, P.S. Alzheimer's Disease Cooperative Study Steering Committee, Siemers, E.; Sethuraman, G.; Mohs, R.; Semagacestat Study Group.A phase 3 trial of semagacestat for treatment of Alzheimer’s disease. N Engl J Med.2013, 369,341–50. Coric, V.; Salloway, S.; van Dyck, C.H.; Dubois, B.; Andreasen, N.; Brody, M.; Curtis, C.; Soininen, H.; Thein, S.; Shiovitz, T.; Pilcher, G. Ferris, S.; Colby, S.; Kerselaers, W.; Dockens, R.; Soares, H.; Kaplita, S.; Luo, F.; Pachai, C.; Bracoud, L.; Mintun, M.; Grill, J.D.; Marek, K.; Seibyl, J.; Cedarbaum, J.M.;  Albright, C.; Feldman, H.H.; Berman, R.M.Targeting Prodromal Alzheimer Disease With Avagacestat: A Randomized Clinical Trial.JAMA Neurol.2015,72(11),1324-33. Muntimadugu, E.; Dhommati, R.;Jain, A.; Challa, V.G.; Shaheen, M.; Khan, W. Intranasal delivery of nanoparticle encapsulated tarenflurbil: A potential brain targeting strategy for Alzheimer's disease. Eur J Pharm Sci. 2016,20 (9), 92:224-34.   Egan, M.F.; Kost, J.; Voss, T.; Mukai, Y.;Aisen, P.S.; Cummings, J.L.; Tariot, P.N.; Vellas, B.; van Dyck, C.H.; Boada, M.; et al. Randomized Trial of Verubecestat for Prodromal Alzheimer's Disease. N Engl J Med. 2019, 11, 380(15):1408-1420. Henley, D.; Raghavan, N.; Sperling, R.; Aisen, P.; Raman, R.; Romano, G. Preliminary Results of a Trial of Atabecestatin Preclinical Alzheimer's Disease. N Engl J Med. 2019;11, 380(15):1483-1485. Vandenberghe, R.; Rinne, J.O.; Boada, M.; Katayama, S.; Scheltens, P.;Vellas, B.; Tuchman, M.; Gass, A.; Fiebach, J.B.; Hill, D.; et al. Bapineuzumab 3000 and 3001 Clinical Study Investigators.Bapineuzumab for mild to moderate Alzheimer’s disease in two global, randomized, phase 3 trials. Alzheimers Res Ther. 2016, 8: The Lancet Neurology. Solanezumab: too late in mild Alzheimer’s disease? Lancet Neurol.2017,16, Scwarz, A.J.; Sundell, K.L.;Charil, A.;  Case, M.G.; Jaeger, R.K.; Scott, D.; Bracoud, L.; Oh, J.;, Suhy, J.; Pontecorvo, M.J.; et al. Magnetic resonance imaging measures of brain atrophy from the EXPEDITION3 trial in mild Alzheimer's disease.Alzheimers Dement (N Y). 2019, 30,5:328-337. Relkin,N.R.;Thomas,R.G.; Rissman,R.A.;Brewer, J.B.; Rafii, M.S.; vanDyck, C.H.; Jack, C.R.; Sano, M.; Knopman, D.S.; Raman, R.; etal.A phase 3 trial of IV immunoglobulin for Alzheimer disease. Neurology.2017, 2, 88(18): 1768–1775. Wilcock, G.K.;Gauthier, S.; Frisoni, G.B.; Jia, J.; Hardlund, J.H.; Moebius, H.J.; Bentham, P.; Kook, K.A.; Schelter, B.O.; Wischik, D.J.; etal.Potential of Low Dose Leuco-Methylthioninium Bis(Hydromethanesulphonate)(LMTM) Monotherapy for Treatment of Mild Alzheimer's Disease: Cohort Analysis as Modified Primary Outcome in a Phase III Clinical Trial. J Alzheimers Dis. 2018,61(1):435-457.

We also added the new information about aducanumab, announced by Biogen 15 days ago. We used dark blue, bold, underlined fonts in the methodological issues and discussion section.

 b. Methodological issues

The most impressive and influential case of correction of a methodological issue is that of aducanumab trials [82]. Aducanumab is an anti-Aβoligomers monoclonal antibody which has been studied in two Phase 3 efficacy trials: The 221AD301 ENGAGE study with 1,350 people with MCI due to AD or mild AD and the identical 221AD302 EMERGE study conducted in 1,350 additional patients. An interim analysis predicted that EMERGE and ENGAGE would miss their primary endpoints, thus aducanumab was removed from any further study. 

On October 22, 2019, it was announced that the interim futility analysis of aducanumab studies was wrong. On the contrary, the subsequent analysis of a larger data showed that EMERGE had met its primary endpoint. It means, that patients receiving the highest dose of 10 mg/kg, had a significant reduction in decline on CDR-SB,  the primary endpoint. The same group also showed a lower decline on secondary endpoints (ADCS-ADL-MCI, MMSE, ADAS-Cog). Subsequently, an exploratory analysis was conducted in the ENGAGE trial which also did not meet the primary endpoint and suggested that the same subgroup of participants declined less and more slowly.

An application for regulatory approval for aducanumab in the FDA is planned for early 2020 [83].

Discussion - Conclusion

The last call for the amyloid hypothesis are probably the monoclonal antibodies directed against Aβoligomers. According to the updated version of the Aβcascade hypothesis of AD, oligomers represent the major pathogenic species of Aβ. Aducanumab (BIIB037) and  BAN2401 (mAb158), anti-Aβoligomers monoclonal antibodies, have shown positive signals of clinical efficacy [85].

Reviewer 4 Report

All comments are answered and the quality of the manuscript is increased significantly. Authors added the requested information.

The manuscript can be accepted for t publication.

Author Response

Thank you very much for the support, time and concern.